# Computational Approaches and Challenges to Developing Universal Influenza Vaccines

**DOI:** 10.3390/vaccines7020045

**Published:** 2019-05-28

**Authors:** Xueting Qiu, Venkata R. Duvvuri, Justin Bahl

**Affiliations:** 1Center for Ecology of Infectious Diseases, Department of Infectious Diseases, College of Veterinary Medicine, University of Georgia, Athens, GA 30602, USA; xueting.qiu@uga.edu (X.Q.); venkata.duvvuri@uga.edu (V.R.D.); 2Department of Epidemiology and Biostatistics, College of Public Health, University of Georgia, Athens, GA 30606, USA; 3Duke-NUS Graduate Medical School, Singapore 169857, Singapore

**Keywords:** Universal influenza vaccine, computational design, interactions of virus-host-environment

## Abstract

The traditional design of effective vaccines for rapidly-evolving pathogens, such as influenza A virus, has failed to provide broad spectrum and long-lasting protection. With low cost whole genome sequencing technology and powerful computing capabilities, novel computational approaches have demonstrated the potential to facilitate the design of a universal influenza vaccine. However, few studies have integrated computational optimization in the design and discovery of new vaccines. Understanding the potential of computational vaccine design is necessary before these approaches can be implemented on a broad scale. This review summarizes some promising computational approaches under current development, including computationally optimized broadly reactive antigens with consensus sequences, phylogenetic model-based ancestral sequence reconstruction, and immunomics to compute conserved cross-reactive T-cell epitopes. Interactions between virus-host-environment determine the evolvability of the influenza population. We propose that with the development of novel technologies that allow the integration of data sources such as protein structural modeling, host antibody repertoire analysis and advanced phylodynamic modeling, computational approaches will be crucial for the development of a long-lasting universal influenza vaccine. Taken together, computational approaches are powerful and promising tools for the development of a universal influenza vaccine with durable and broad protection.

## 1. Introduction

In the history of fighting infectious diseases, vaccinations are amongst the most cost-effective approaches available to prevent infection. Traditional approaches to vaccine design have been successful against many pathogens. But vaccines that target rapidly-evolving and genetically-diverse disease agents have frequently failed to generate long lasting protection for human populations. This is particularly true for influenza viruses, a single-stranded, negative sense RNA virus. One of the important weapons being developed to effectively prevent influenza virus infection is a vaccine that can provide durable and broadly-reactive protection against multiple subtypes, including those that may cause potential pandemics, that is, a universal influenza vaccine [1]. The National Institute of Allergy and Infectious Diseases (NIAID) has defined the criteria for universal influenza vaccine, which includes (1) being at least 75% effective against symptomatic influenza infection; (2) protecting against group I and group II influenza A viruses (influenza B would be a secondary target); (3) having durable protection that lasts at least 1 year and preferably through multiple seasons [1]. These are challenging but achievable goals to effectively develop a vaccine that can protect against the globally-disseminated virus. 

Recent approaches to vaccine design have taken advantage of large-scale viral sequencing platforms, phylogenetic frameworks, protein structural modeling and systems biology to design novel broadly-reactive vaccine candidates, which have been used for influenza and other pathogens [2]. These new approaches have revealed insights of viral evolution, transmission dynamics and biological functions of proteins from mountains of genomic data and metadata [2,3,4,5]. Novel approaches for rational design in the genomic era can aid in achieving goals of universal influenza vaccine design. However, it has found limited applications in the design and discovery of new vaccines, an area where proper integration of computational support and design is urgently needed [2,6]. 

In this review, we aim to briefly summarize the currently applied approaches of seasonal and universal influenza vaccine design and their disadvantages (part 2), gather information on new or potential computational approaches and challenges (part 3), and to propose necessary resources and efforts needed for computational approaches of universal influenza vaccine candidates (part 4). We will explore the important role of computational vaccine design to improve the identification of pathogen antigens and key components for designing and evaluating a universal vaccine design. Furthermore, we will discuss the potential of incorporating interactions of virus-host-environment to develop models that allow for precise prediction for viral evolution and vaccine candidates. This review provides a framework to integrate computational advances that could help in restructuring the existing seasonal influenza vaccine design and contribute to the development of universal influenza vaccine.

## 2. Current Approach for Influenza Vaccine Design

### 2.1. Selection of Circulating Influenza Viruses for Seasonal Vaccine Design

To prevent infections from circulating seasonal influenza viruses, the annually administered influenza virus vaccines contain H1N1 (phylogenetic group 1 hemagglutinin), H3N2 (phylogenetic group 2 hemagglutinin) and two influenza B virus components (Victoria-like and Yamagata-like) [7]. The vaccine candidates from natural influenza virus strains are recommended by the World Health Organization (WHO) based on the characterization and prediction of circulating strains likely to dominate in upcoming epidemic seasons. Twice a year, the expert panel from the WHO Collaborating Centers and essential laboratories and academies reviews the evidence of global surveillance, laboratory and clinical studies and evaluate the availability of vaccine strains to make recommendation on the components of influenza vaccine [8]. The evaluations are mainly based on viral antigenic and genetic characterization, which requires tremendous annual surveillance efforts and laboratory tests. After the selection of vaccine strains, it takes at least 6-8 months to produce sufficient global supplies of influenza vaccine via current vaccine production technologies with egg-based, cell-based or recombination-based vaccine [9,10]. For a comprehensive review of traditional approaches for influenza vaccine selection, design, development and challenges refer to this review paper by Wong and Webby [11].

Influenza vaccines selected from natural influenza virus strains predominantly elicit specific antibodies against the globular head domain of the surface protein hemagglutinin (HA) for each subtype or lineage, which is only effective to protect against closely-matched antigenic variants [7]. The HA, however, undergoes rapid antigenic drift that accumulates from point mutations under immune selection pressure in the major antigenic sites, allowing the virus to escape neutralizing antibody responses [12], resulting in imprecise predictions of circulating strains. Despite significant efforts of continuous surveillance and vaccine strain updates, vaccine mismatches have occurred many times [13]. In addition to potential antigenic mismatch from selection procedure and delays in production, egg-adapted mutations accumulated during egg-based vaccine production can further exacerbate this issue, where the vaccine virus strain obtains relevant functional amino acid changes in the HA protein, resulting in low vaccine effectiveness [14,15,16,17]. Studies investigating the impact of vaccine mismatch have reported broad ranging vaccine efficacy (10% to 60%) for these annual vaccines, demonstrating severely low and unstable immune protection from influenza infection [18]. Predictive models of viral evolution to forecast dominant circulating influenza viral strains in the upcoming influenza seasons through the analysis of genetic and epidemiological data from influenza surveillance system have been developed to make quantitative predictions of viral evolution and aim to improve the selection of seasonal influenza vaccine candidates [10,19]. This framework has demonstrated potential to integrate multiple data sources to improve influenza vaccine design.

### 2.2. Universal Influenza Vaccine Design

Seasonal vaccines offer a little or no protection to emerging zoonotic influenza viruses with pandemic potential, as many species, especially wild aquatic birds, are recognized as the natural reservoir of all subtypes of influenza A viruses and have the potential to occur spillover and infect humans directly [20]. As with past pandemics, the surface glycoproteins, HA and neuraminidase (NA) are replaced through reassortments of zoonotic strains where the human population has no pre-existing immune protection and the vaccines in use are not cross-reactive with these new strains [21,22,23]. Experimentally-identified, conserved and immunogenic M2 protein antigens [24], and HA-stalk design [24,25,26] have the potential to elicit broadly protective antibodies against seasonal influenza strain. M2-based universal vaccine design focuses on the conserved antigens that have been experimentally identified on M2 protein. However, the low immunogenicity and epitope density by viral nature has been a fatal limit to make the cross-protection from M2 being effectively applied into vaccine design [24]. To solve this issue, many approaches have been developed to improve M2 immunogenicity, details of which can be found in this review by Zhang et al. [24]. Similar with M2-based design, HA-stalk design tries to elicit the conserved and cross-reactive protection from the membrane-proximal stalk domain [25]. While the stalk domain is conserved across multiple influenza subtypes, it is shielded by the immune-dominant head domain. To amplify the broad protection from stalk domain, truncated HA without head domain, concentrated short peptides from stalk domain or recombinant chimeric HA proteins have been employed [24,25,26]. Despite the potential for both M2 and HA-stalk design vaccines to elicit broadly reactive immune response, a number of challenges remain (reviewed in [24] and [27]), including a limited understanding of the full repertoire of potential epitopes. More systematic computational approaches that go beyond circulating strain prediction and incorporate a full profile of antigens stimulating both humoral and cellular immune responses are needed for universal vaccine design [24,25,26]. To overcome these challenges, computational approaches have been employed to rationally design promising vaccine candidates that can induce broadly (ideally universally) cross-protective and durable immunity for all seasonal and even emerging pre-pandemic strains [13,28,29]. 

## 3. Computational Design of Universal Influenza Vaccines 

### 3.1. The Rationale of Computational Design Approaches 

Traditional approaches have failed to produce stable and protective vaccines for hypervariable and rapidly-evolving viral pathogens, including influenza viruses [30,31]. The reasons for this failure include inherent uncertainty in pathogen evolution [32]. While global surveillance efforts and data sharing agreements have increased available information, vaccine design often ignores the underlying processes of the global influenza meta-population which generates diversity that allows the viral populations to escape vaccine-induced immune responses and anti-viral treatments. Furthermore, hemagglutination inhibition assay, central to vaccine strain selection, is a poor approximation for the average immune response that does not account for the heterogeneity of immune responses between hosts and pathogens, which cannot provide a full profile of pathogen immunogenic features [33]. The failure to synthesize information across the host-pathogen-environment, including ecological and epidemiological determinants of disease persistence and spread (Figure 1) [2], has resulted in major information gaps that can be addressed by existing computational approaches and a concerted effort to develop a unified framework. Individual immune response to a vaccine is an interplay of genetic, molecular and ecological factors from both host and pathogen populations on large tempo-spatial scales [2]. As a consequence, traditional design inefficiently captures few pathogen features based on a limited input that does not account for the high diversity of pathogen and high heterogeneity of host’s immune responses [2,34]. 

Fortunately, the growth of databases containing genome sequences sampled throughout global epidemics [35,36,37], increased computational power and theoretical algorithms allow complex data sources to be integrated into a unified framework allowing for a more complete understanding of pathogen and host features. Huge amount of data generated by the high-throughput technologies are currently available with more data regularly being made available. Computational approaches with advance data integration and quantitative empirical analyses fit the needs of universal vaccine design for highly diverse influenza virus in several promising aspects [38,39]: (1) being able to model and analyze all available viral genomic data over a large tempo-spatial scale and shift from HA only design to cover more antigens on multiple viral proteins; (2) rapidly and cost-effectively screening antigens and epitopes in the early phase of vaccine candidate discovery; (3) capability of incorporating protein functional structure and antibody repertoire analysis via structural biology; (4) machine learning to incorporate viral, ecological, epidemiological and host immunological data to make precise assessment and prediction.

Computational approaches to identify candidates for universal influenza vaccine design have been used with a variety of novel vaccine production strategies in development. These approaches mainly focus on the ‘unnatural immunity’ [40] induced by more conserved or less immune-dominant domains in the surface proteins, internal proteins or both, to tackle with the high degree of variability in influenza viruses by boosting the immunity from the conserved or less evolvable proteins of the viruses. Current rational vaccine design uses comparative genomic methods to identify these conserved regions. These inferential methods include naïve approaches where conserved regions are identified from multiple sequence alignment comparison [27,28,41,42,43], phylogenetic approaches where common ancestry is estimated [44] and peptide engineering based on 3-D protein structure and immunomics. Figure 2 has summarized these current computational approaches. Table 1 has highlighted the advantages, disadvantages and examples of these approaches. 

### 3.2. The Host

#### 3.2.1. Immunoinformatics to Immunomics

The field of immunoinformatics or computational immunology received major attention in 2000’s from the research and governmental funding agencies [49,50]. Immunoinformatics research mainly focuses on study and design of high-throughput in-silico approaches to explore the immune system at genome level (Figure 2) [51]. These technological developments coupled with pathogen genomes have tremendously contributed to the selection process of optimal vaccine antigens by lessening the time and cost involved in the conventional methodologies that involve pathogen cultivation and protein extractions. This methodology of analyzing pathogen genome to identify potential vaccine antigens is called “reverse vaccinology” [52,53]. 

The study of immunomes coined as a new discipline “immunomics”, where the ‘immunome’ is quoted as “the detailed map of immune reactions of a given host interacting with a foreign antigen” [50]. Immunomics tools such as B-cell epitope and T-cell epitope mapping methods mimic the diverse molecular pathways of adaptive immune system that accounts for humoral immunity (B-cells) and cellular immunity (CD4+ T-cells and CD8+ T-cells) to predict potential epitopes or immunomes from the pathogen proteomes [51,54]. B-cell epitopes are surface exposed clusters of amino acids, which can be categorized as linear (a stretch of amino acids) and conformational (discontinuous) epitopes recognized by B-cell receptors (BCR) [55]; while T-cell epitopes are only linear, and T-cells receptors (TCR) can recognize epitopes when they are bound to the major histocompatibility complex (MHC) molecules. Two distinct subsets, CD4+ T-cells (helper T cells) and CD8+ T-cells (cytotoxic T cells) recognize epitopes when they bind with MHC class II and MHC class I, respectively [56]. MHC genes are highly polymorphic across different ethnicities that determines the fate of an epitope presentation to T-cells [56].

Immunomics can aid in identifying optimal B-cell and T-cell epitopes directly from the pathogen proteomes, while the literature suggested that T-cell predictions are more advanced and reliable than that of B-cell epitope predictions [57,58]. A workshop on the B-cell epitope prediction tools reported that the prediction performance of B-cell tools is still far from reality due to a lack of high-quality experimental datasets [57]. Detailed description on the existing epitope mapping tools, and challenges have been discussed in the cited review articles [55,58,59,60,61,62]. Key limitations include: (1) the availability of experimental datasets essential in training and developing any epitope prediction tool; (2) selection of epitope prediction tools may also introduce discrepancy in the identification of potential T-cell epitopes due to methodological differences. T-cell epitope prediction tools that include sequence- and structure-based methods are reviewed in Patronov et al. [56] and Luo et al. [63]); (3) the availability of high-quality datasets on the binding affinity of epitope-MHC, which directs the development and success of T-cell epitope prediction tools. A prediction of strong binding affinity suggests that an epitope will be presented to T-cells. But this requires an experimental assessment; and (4) The population coverage of an epitope is related to MHC polymorphism that exists in humans. The efficacy of epitope-based vaccine(s) can be limited due to variability of MHC alleles among different ethnicities. This may reduce the maximum population coverage of epitope-based vaccine leading to the failure of the vaccine to elicit T-cell immune responses. The current tools, IEDB population coverage [64] and EPISOPT [65], that are used to predict the population coverage are based on the limited experimental HLA frequency data from world-wide MHC allele frequency database (http://www.allelefrequencies.net/).

T-cell immunity plays a critical role in viral clearance thereby reduction in disease severity. Particularly, memory CD4+ T-cells can provide substantial protection against influenza infection through direct effector mechanisms as well as indirect regulatory and helper functions [66,67,68]. In the absence of neutralizing antibodies, the cross-reactive T-cell immune responses towards the well-conserved T-cell epitopes may play a significant role in promoting clearance of virus and reducing disease severity [69,70,71]. This phenomenon was well documented during the 2009 pandemic H1N1, as its unanticipated milder disease severity was largely attributed to the preexisting cross-reactive T-cell immune responses towards the evolutionarily conserved T-cell epitopes between seasonal H1N1 and 2009 H1N1 strains [72,73,74,75,76,77,78]. Taken together, these studies suggest that an epitope-based universal influenza vaccine can be developed by selecting the well-conserved and immunodominant epitopes across influenza subtypes using the immunomics approach.

A major challenge in the design of epitope-based vaccines is to focus immune response onto multiple well-conserved epitopes in order to elicit broad protective/neutralizing immune responses. Epitope grafting or scaffolding, has been proposed as a solution for epitope-based vaccine design. In this method, minimal epitopes that are highly conserved in pathogen are grafted onto an appropriate heterologous-protein scaffold. Approaches for scaffold selection and design include single algorithm-based tools like MAMMOTH or meta-servers like TM-align and consensus-based designs [30]. Three main criteria have been proposed for the selection of scaffold that include size, where smaller-sized scaffolds help to focus immune responses to grafted epitopes while preventing unwanted responses to scaffold. Second criterion is the flexibility of scaffold with a possible positive correlation between flexibility and immunogenicity. The third criterion is the structural environment of the graft. A well-defined structural boundary between protein scaffold and epitopes enhances the specificity of immune responses [30].

#### 3.2.2. Advanced Universal Influenza Vaccines in Clinical Development

There are currently three promising epitope-based universal influenza vaccines, FP-01.1, Flu-v and Multimeric-001 (M-001) are at different stages of clinical trials (Table 2). Each vaccine is briefly described below.

FP-01.1 vaccine (also called as Flunisyn™), comprises six different synthetic peptides (length: 35 amino acids) each conjugated to the fluorocarbon moiety C8F17(CH2)2-COOH. These epitopes were derived from the nucleoprotein (NP), matrix protein (M), and polymerase basic proteins (PB1 and PB2) and have high level conservancy across H1-H9 influenza A subtypes with wider population coverage. The phase I clinical trial [79] results observed that vaccine has acceptable safety and tolerable profiles and generate robust CD4+ and CD8+ T-cellular immunity [80]. 

Flu-v vaccine contains multiple highly conserved T cell epitopes derived from NP, matrix proteins (M1 and M2) from influenza A and NP from influenza B viruses and are conserved across most influenza viruses with high population coverage [81,82]. The phase II clinical trials with adjuvant+Flu-v triggered the T-cellular responses and also induced antibody response [83].

Multimeric-001 (M-001), a universal influenza epitope-based vaccine, is currently at the pivotal phase III clinical trial to assess the safety and clinical efficacy as a standalone universal flu vaccine in participants with age of older than 50 for a two-year follow-up [84]. M-001 comprised with a single recombinant protein that contains nine linear, conserved and common epitopes from NP, M1, and HA of influenza A and B viruses to activate both humoral and cellular immune system to provide multi-strain protection from the seasonal and pandemic influenza viruses [85]. The predicted population coverage of these selected epitopes is greater than 90%. The epitopes from the HA1 region which is hypervariable were not included in the M-001. At phase II clinical trial in 120 participants aged 65 years and older, M-001 was first administered to the study participants and three weeks later they were immunized with 2011-2012 seasonal trivalent inactivated vaccine. Results reported that M-001 alone elicited cellular responses and enhanced HA inhibition seroconversion to 2011/12 vaccine strains, and even to certain former vaccine strains [86]. 

The positive note on the epitope-based universal vaccine efficacy in eliciting the robust immune responses at clinical trials underpins the immunomics in advancing the current vaccine development approaches to prevent infections from remerging or emerging highly evolving influenza viruses.

#### 3.2.3. Computational Approaches that Incorporate Host Immunological Factors

Antibody repertoire analysis can be used to directly incorporate host immune response for vaccine design. It combines sequence analysis with structural modeling and machine learning to predict and analyze all the antibody affinity and specificities that can be produced by an individual, which can be a valuable tool for quantitative evaluation of vaccine-induced immune responses [30]. Though it is currently used to characterize broadly neutralizing antibody (bnAb) lineages, with the development of next-generation sequencing (NGS) technologies and systems biology, the analysis of antibody repertoire encoded by B cells in the blood or lymphoid organs can be used to understand humoral immune responses and to identify antibodies specific for antigens of interest in animal models and human vaccine trials [30,87,88,89]. The antibody NGS can have impact on the rational vaccine design by decoding the human immune responses and delineating B and T cell antigen receptors [90,91]. This approach has been well developed in HIV-1 to identify hypervariants and evolution on neutralization and binding to bnAbs [92,93] and explore the antibody lineage via phylogenetic modeling [89,94]. These technologies and bioinformatics tools can be applied to influenza virus vaccine design with creating library of antibody repertoire by NGS. The library then can be used in computational approaches to quantitatively measure the immune responses and further to predict the effects of vaccine candidates without completely relying on costly animal tests. 

The main limitation with this approach is that linear sequence may not accurately predict the conformational variations when these antigens are put back in a complete protein context [95]. When the conformational structure of the epitope is not accurate, the corresponding immune response cannot be precisely computed [96]. To solve this issue, some high-performance bioinformatics tools such as molecular dynamics simulations can be used to predict the 3-D structure and stability of proteins or peptides [97,98]. Furthermore, in the previous section, the successful maintenance of the conformational epitope in these clinical tested vaccines has provided positive evidence for epitope-based universal vaccine design. Taken together, with this antibody repertoire analysis tool, the computational estimation of immune stimulation of these predicted viral antigens in hosts could be more accurate. 

### 3.3. The Pathogen

#### 3.3.1. Model-free Consensus-Based Optimized Approach

Consensus sequences are usually generated by aligning and comparing multiple sequences and selecting the most common residue at each position (Figure 2A). These sequences are expected to effectively capture a profile of conserved genetic and epitope information which can induce cross-reactive cellular immune responses [99]. The outcome of this approach is a sequence alignment with conserved antigens that can be expressed on virus-like particle (VLP), which are similar to intact virions but not pathogenic [41,100]. Influenza VLP vaccines have advantages that a live virus is not used at any step during vaccine production [101] and they can maintain conformational epitopes by presenting surface antigens in their original structures. Consensus-based studies [29,99,101] have generated consensus sequences for NA protein of H1N1 and several influenza proteins of H5N1, including HA, NA and matrix protein M1, which have elicited broadly-reactive immune response. However, the nature of consensus-based antigen design determines that it is highly influenced by the input sequences and thus subject to sampling bias [102]. For example, H5N1 isolates were sampled in different geographical locations and from different hosts, including human and avian. If samples from one location or one host are overrepresented in the sequences used to generate consensus, then it can bias the output consensus sequence, which may not accurately represent the full conserved genetic profile of the whole H5N1 population. To overcome issues from sampling bias, an iterative optimization strategy has been implemented in an approach known as computationally optimized broadly reactive antigens (COBRA) [41]. 

The critical step in designing COBRAs is to use multiple rounds of consensus generation. Within each phylogenetic subclade of the influenza virus subtype, the primary consensus with the most common amino acid at each position is generated for each individual outbreak group that is defined based on geographic location and collection time. The secondary consensus is generated from the primary consensus to represent the subclade. The third or fourth consensus is generated based on previous round of consensus, until the final consensus is generated and termed COBRAs [41]. 

The COBRAs generated by multiple rounds of consensus generation are representative of the diversity in the viral population and are able to induce neutralizing antibodies or other immune boosting response to protect against past, current and ideally, future circulating strains of this specific HA subtype [27]. COBRAs-based designed HA protein of H1, H3 and H5 have been tested with *in-vitro* assays and animal models. This preclinical evidence has showed broad HA inhibition antibody titers that were cross-reactive with different strains within the same subtype [28,41,42,43,45]. This approach has advantages over other universal vaccine candidates, because COBRA HA-elicited antibodies are able to neutralize the receptor binding site and the design has a clear path towards clinical proof of correlate for protective efficacy in humans [28]. However, there are some concerns associated with this approach. To be universally cross-reactive, the ideal CORBA HA protein should contain all the conserved information present in multiple subtypes. The conserved immunogenic profile of consensus sequences from COBRA approach is dependent upon the sharing of epidemiological and genetic data collected during public health investigations and surveillance of outbreaks. With biased viral samples, the consensus sequence generated may not represent the full profile of conserved immunogenicity along viral evolving history. Even with increased global efforts to collect data and characterize epidemics it is unlikely that sufficient data could be collected to overcome this challenge. Alternative approaches, such as phylogenetic modeling of viral proteins along a characterized evolutionary trajectory that account for impacts of sample biases and missing data could greatly improve design of COBRA candidates.

#### 3.3.2. Phylogenetic Model-Based Approaches to Ancestral Sequence Reconstruction 

Another way to identify potential broadly reactive antigens is ancestral sequence reconstruction, which is to computationally infer ancestral gene sequences and the translated ancestral protein sequence (Figure 2B) [103]. Ancestral sequences can reveal conserved functions of the pathogen protein where the potential cross reactivity of the ancestral virus would also be evolutionarily conserved [104]. These conserved functions may indicate potential immune targets. Phylogenetic evolutionary models have been used to infer influenza viral evolutionary history for decades with molecular data, including the analysis of large phylogenetic trees, complex evolutionary models for more accurate ancestral inference and detection of the imprints of selection pressure in molecular sequences [105,106]. Phylogenetic algorithms have been developed to reconstruct ancestral sequences for broadly-reactive vaccine design [44,107]. This phylogenetic approach with marginal reconstruction yields the maximum likelihood at the site with a specific amino acid after comparing all probabilities of different amino acids at a site on an internal node [107]. It can more accurately account for sampling bias and the variability of substitution rates among sites that can affect the consensus approaches described above. 

In detail, Ducatez and colleagues [44] developed an ancestral sequence reconstruction method for highly pathogenic avian influenza (HPAI) H5N1 surface proteins HA and NA. Based on a maximum likelihood tree, several ancestral sequences were reconstructed at the internal nodes of co-circulating HPAI H5N1 viral lineages to capture the conserved genetic characteristics of these viruses. These ancestral sequences were synthesized into attenuated influenza viruses that could replicate. Their cross-reactive protection against H5N1 morbidity and mortality have been confirmed in preclinical experiments with ferret models. These findings provide strong evidence that computationally derived vaccine candidate sequences and these technologies should be used to explore and enhance the cross-reactivity, which can be easily fit into the current licensed vaccine platform. These computationally derived ancestral sequences as vaccine candidates may help in avoiding the detrimental effects of antigenic drift on the vaccine effectiveness. But this approach can be weakened by phylogenetic uncertainty in particular when trees possess long branches due to insufficient information [108]. 

The functional and structural domains of pathogen protein can be under disparate immunologic pressures and thus have impacts on the evolutionary phylogeny [109] and the accuracy of ancestral sequence reconstruction. Even though advanced models, including those that account for protein sequence and structure [110,111] have not been applied for vaccine design, the computational approach is promising. Precise estimation of influenza virus evolution including protein structural and its functional information supported by experimental data [112], may help to efficiently identify and select target antigens for universal vaccine design [30]. 

The integration of protein functions and structures into evolutionary models has two main challenges: (1) published viral protein structural and functional information may not be available or sufficiently resolved based on current studies; and (2) The assumption of nucleotide site independence in the model cannot capture the biological reality that some sites are linked due to shared function [113]. Some modeling approaches with a protein structure scoring system or partitioning schemes on the protein sequence [98,110,111] can potentially overcome these challenges; for example, protein structure has been explored with coarse-grained models for structure prediction, prediction of protein interaction and molecular dynamics simulations of protein folding [98]. This provides the statistical potential like a scoring system for sequence-structure compatibility, which can be used to evaluate the probability of fixation of a given mutation and improve the precision of ancestral reconstruction [111]. However, few studies have incorporated protein structural information into the evolutionary analyses. Simple representations of protein functional and structural domains have been used so far. Hypothetically, novel models with a more complete representation with a full site mapping of the protein functions and structures would yield a better fit. But in a phylogenetic context, structurally informed models are still outperformed by some site-independent models in terms of fit [111]. Preliminary data suggest that this would become less of a concern with increased sharing of sequence data [110]. 

High-throughput experiments quantify the effects of all single substitutions on gene function so that evolutionary model can adequately capture the heterogeneity of selection at different sites, which may improve phylogenetic inference and ancestral sequence reconstruction [112,114]. The new experimental technique is called deep mutational scanning, where a gene is randomly mutagenized and subjected to functional selection in the laboratory, and then deep sequenced to quantify the relative frequencies of mutations before and after selection [115,116]. This technique has been used to quantify the impacts of codon changes to several proteins or functional domains [114,115,117,118,119,120]. This information of protein function from rapid high-throughput experiments may greatly improve the precision of ancestral sequence reconstruction [121].

### 3.4. The Environment

#### Pathogen Evolvability 

Uncovering the important ecological, immunological and environmental determinants on viral evolution is very important to make predictions of the viral emergence, fitness, transmissions and circulating potential after new substitution is introduced [122]. Evolvability, first coined by Kirschner and Gerhart in 1998, means that the organism’s capacity to generate heritable phenotype [123]. The zoonotic nature and complicated ecology of influenza viruses make evolvability more difficult to quantify and predict. But with the advances of phylogenetic algorithms, models can integrate and evaluate the impacts of environmental determinants. For example, an important development in phylogenetic modeling was the field of viral phylodynamics that was introduced in 2004 to study “how epidemiological, immunological, and evolutionary processes act and potentially interact to shape viral phylogenies” [124,125]. Dynamics of influenza virus infections and transmissions at individual-level (such as viral evolution within an infected host), population-level (individual hosts within a population), or ecology-level (entire populations of different host species) have been studied [124]. Specially, phylodynamics have been used to study factors of interest on some viral phenotypes, including virulence, viral transmissibility, cell or tissue tropism, and antigenic phenotypes that can facilitate immune escape, etc. [108,124]. Details of methods and examined significant factors can be found in these reviews [108,124,125].

Furthermore, the complements between phylodynamic modeling and experimental testing can be integrated together to improve prediction on influenza virus evolvability. For example, experimental studies designed to assess viral evolvability [126,127] demonstrated that a measured fitness score or estimated tolerance for mutations can be used in phylodynamic modeling to link phenotypes, genetic characteristics and other ecological factors, which can improve the prediction of viral evolvability for natural influenza virus strains [127]. The potential predictors and consequent mutations computed by models can enhance our understanding on viral characteristics, potential immune escape, or influenza antiviral drug resistance [128]. Challenges for this area are how to get accurate and sufficient information on the epidemiological, immunological and ecological factors, how to expand, integrate and enhance phylodynamic models [108], and how to gather the current modeling factors to improve prediction of viral evolvability [122].

## 4. Resources and Efforts Needed for Computational Vaccine Design 

Computational models with incorporating host-pathogen-environment can efficiently facilitate the understanding of viral evolution and the selection on critical information for vaccine design. With the challenges summarized above, extra resources and efforts are needed for developing computational vaccine design. 

### 4.1. Data Collection and Sampling Efforts 

Computational vaccine design relies on the input data quality [30]. To be specific, the representative of the collected samples, the completion and precision of recorded data, and the timely manner of data sharing and availability can ameliorate the output from computational modeling [19,129,130]. Compared to other infectious disease sampling, influenza viruses have already established an excellent global network of sentinel institutions to monitor outbreaks and collect human samples [131]. With the lower cost of full-genome sequencing, a large amount of genetic data has been available for influenza research. However, three main limitations exist in current surveillance: (1) the imbalanced sampling efforts on different hosts and geographical regions; (2) the incompletion of data records [129]; and (3) the delayed availability of sequence data [19,130].

The unequal sampling of geographical regions is caused by global and local resource allocation [132]. Policies to globally optimize resource allocation with considering the representative of collected samples from outbreaks in different regions are needed. But majorly, the unequal sampling in zoonotic hosts is more severe. Human influenza outbreaks have been well monitored and sampled [129]. However, to better understand viral evolvability and predict potential pandemic emerging from zoonotic strains, more sampling efforts are definitely necessary in animal hosts, especially wild aquatic birds [129]. Olson et al. [129], examined 11,870 GenBank records and reviewed 50 non-overlapping studies and over 250,000 birds to access the status of historic sampling efforts during 1977–2012, where they found that sampling in different hosts, location and viral subtypes are severely imbalanced and there are a high proportion of non-tested samples globally. If we aim to identify a high proportion of the virus subtypes in circulation in a given time period with limited resources, a sample-based accumulation curve can provide an initial rationalization and optimal sample size for avian influenza virus surveillance [129,133,134].

The affiliated sequence meta-data records have been improved with samples from recent years; however, the epidemiological information, viral phenotypic characteristics and host characteristics are not sufficiently recorded. With no accurate information on geographical region, host species and migratory pathways and viral characteristics, we do lose lots of power in our model inference [135], not to say improving the prediction of viral evolvability. GISAID [35] and GenBank [36], these open access database platforms have facilitated the accessibility and sharing of influenza sequence data to the science community. Despite the availability of these platforms, the sharing of viral sequence data is often long after the outbreaks and records are frequently incomplete [136]. Therefore, a standardized protocol on how to record collected samples and what information is needed to report should be established for sharing more complete viral and host-related information. 

### 4.2. Integration of Experimental Evidence and Model Development

As shown in Figure 2, computational models that fully integrate multiple sources of information, including experimental evidence, could aid in the identification of critical components for vaccine design. However, we cannot solely rely on computational design, where computed antigens have uncertain biological effects. Experimental evidence (Figure 1) from animal models or approved human clinical trials are valuable to be incorporated into computational design. The experimental data on pathogen immunogenicity and host immune system can first provide preliminary evidence on natural or computed antigens and further amplify the usage of this new evidence to the computational procedure for more accurate prediction and evaluation [19].

More complicated and realistic models previously limited by computing capability can be developed with the advances of computing power [137]. For example, it becomes possible to develop viral phylodynamic models that can incorporate results from laboratory experiments of viral antigens and host immune responses [108]; The development on structured coalescent for better estimation on viral population and mutation or migration events [138,139]. Furthermore, to avoid overparameterization, model selection procedure should be applied during the process of novel model development to optimize the balance of biological reality and parameterization [137,140]. With all these, the next step would be to introduce and apply machine learning to the computational process for vaccine design. 

Machine learning is a subset of artificial intelligence in the field of computer science, which usually uses statistical techniques and mathematical models to make computers “learn” with data without being explicitly programmed, that is, performance on a specific task progressively improves [141]. Machine learning algorithms discover patterns in data and construct mathematical models using these prior discoveries. One advantage of machine learning is that the models can be used to make predictions on future data by cumulating from previous evidence and improving on forecasting algorithms [142,143,144]. Though still in an early phase of implementation, the concept of machine learning has been used in viral evolutionary modeling and has been a rapid way to gather and update information based on known information [145,146]. Machine learning can incorporate different modeling steps and all available surveillance, genetic and experimental data to keep updating information and make predictions for computational vaccine design (Figure 1). For example, the model of conserved epitope prediction mentioned in previous sections can also be incorporated into the platform with host and environmental factors to make prediction on currently circulating viruses, where broad-reactive vaccine candidates can be rapidly computed and tested. 

## 5. Conclusions 

For decades, we have been using the traditional approaches to design and develop influenza vaccines. The rapid genetic changes and antigenic drift of influenza virus populations result in short-term protection necessitating continual vaccine updates with novel viral components based on analysis of globally circulating variants. Furthermore, these vaccines do not offer any immunological protection against potential pandemic zoonotic strains, one of the lessons learned through the unprecedent appearance of swine-origin 2009 pandemic H1N1 virus. 

Recent decades have witnessed the technological advancements in the viral genetic sequencing and computational modeling in tracing the complexities involved in the interactions of host- pathogen-environment that produced important insights into influenza disease dynamics across biological scales. Integrating these computational and technological pipelines into the vaccine design protocols can facilitate the development of a broadly cross-reactive, evolutionarily-resistant universal influenza vaccine. 

## Figures and Tables

**Figure 1 vaccines-07-00045-f001:**
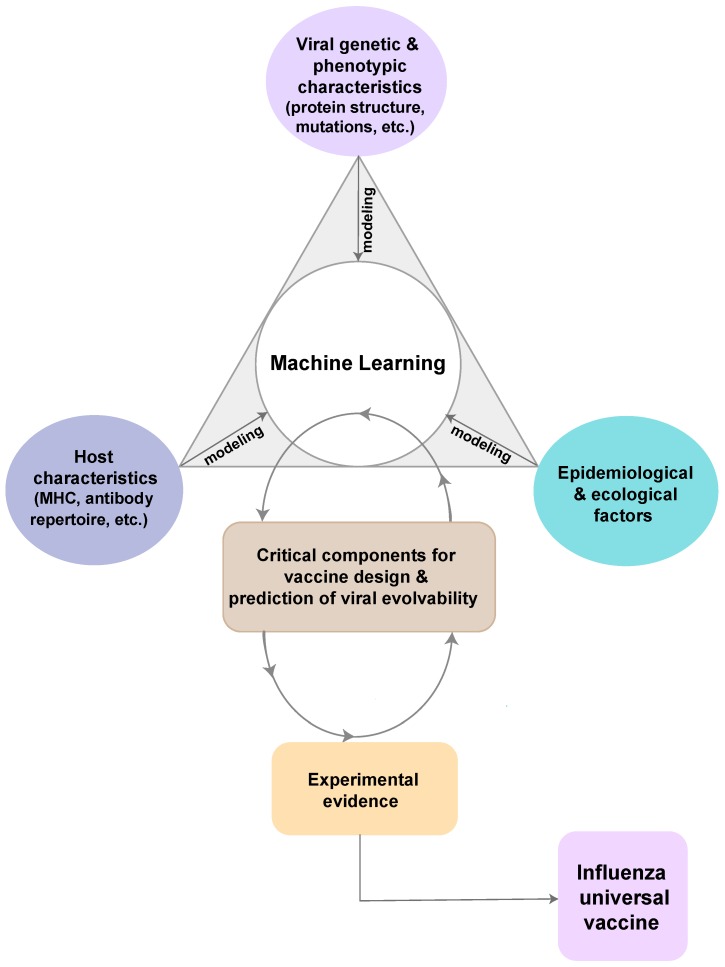
Framework of potential novel computational design. The summarized potential approaches combine the epidemiologic triad of infectious diseases. From host perspective, data on host characteristics are modeled to understand the susceptibility and immune response to influenza viruses by studying the host immunogenetics, for example, the antibody repertoire analysis or the human leukocyte antigen (HLA) structure analysis. From virus perspective, phylogenetic modeling is to understand the evolutionary history and patterns of viral genetic and phenotypic characteristics. Further, developing phylodynamic modeling, generalized linear model (GLM), and other more advanced models are critical to identify important epidemiological and ecological determinants that affect viral evolution and host immunity to influenza virus. In order to generate critical components for vaccine design and accurately predict viral evolvability, all three perspectives are combined to form a machine learning pipeline to incorporate information and learn from these data and models. Evidence from experimental tests on these components can be used into machine learning pipeline to improve outputs. Many iterations are needed with input of more information. The ultimate goal is to generate high-quality information and broad-reactive components for a universal vaccine of influenza viruses.

**Figure 2 vaccines-07-00045-f002:**
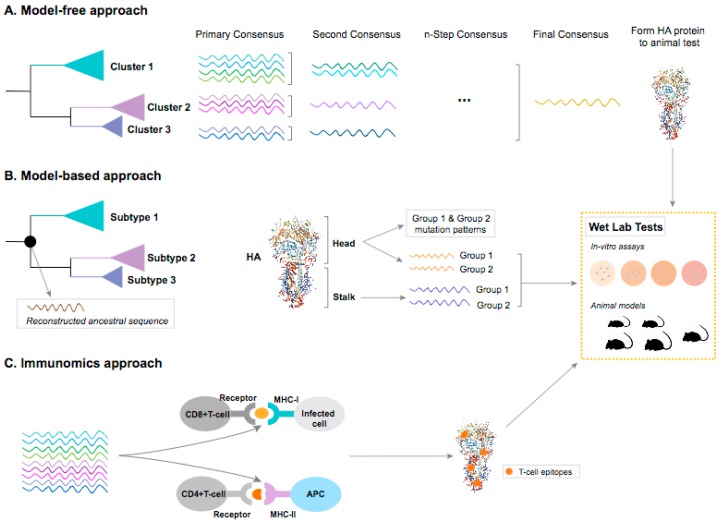
Framework for computational influenza universal vaccine design. (**A**) Model-free consensus-based optimized approach. Consensus sequences from previously defined clusters are generated by aligning and comparing multiple sequences and selecting the most common residue at each position. It may go through several steps until the generation of a final consensus. (**B**) Model-based ancestral sequence reconstruction approach. Maximum likelihood and Bayesian approaches are the most commonly used statistical phylogenetic methods to reconstruct ancestral sequence at the ancestral node (shown as black dot on the tree) [46,47,48]. Evolutionary models that incorporate protein structural domains can be used to separately estimate the evolutionary history on each functional partition as the HA head and stalk domains. Based on the evolutionary relationship among different subtypes of influenza A virus, common ancestral sequences of head and stalk domains can be generated within influenza A virus Group 1 (H1, H2, H5, H6, H7, H8, H9, H11, H12, H13, H16, H17, and H18) and within Group 2 (H3, H4, H7, H10, H14, and H15), respectively. (**C**) Immunomics approach. The T-cell epitope prediction tools can be used to identify the potential CD4+ T-cell and CD8+ T-cell epitopes from the pathogen proteome or protein(s) based on the high binding affinity between epitope-Major Histocompatibility Complex (MHC) complex. Some epitopes will be presented by the MHC-I on the surface of infected cells or by MHC-II on the surface of antigen presenting cells (APC) to the host CD8+ or CD4+ T-cells, respectively. These processes elicit the cellular and/or humoral immunity. The predicted T-cell epitopes that are evolutionarily conserved and common across or within subtypes will be constructed into peptides or proteins. All outputs from these three approaches, like epitopes, peptides, proteins or virus-like particles (VLPs), will be tested at in-vitro and/or in-vivo models to evaluate their immunogenicity. The proposed concept as shown is based on HA gene sequences, but these approaches should be used for all the gene segments of influenza viruses to generate a full profile of viral immunogenicity.

**Table 1 vaccines-07-00045-t001:** Summary and examples of computational influenza universal vaccine design.

Approach	Conceptual Design	Evidence-Level	Advantages	Disadvantages	**Examples**
**Consensus-based optimized approach**	Figure 2A	Pre-clinical	(1) Efficiently generate a potentially full profile of conserved immunogenicity in viral genome; (2) Induce broad HA inhibition antibody titers that are cross-reactive with diverse strains within the same subtype; (3) Neutralize the receptor binding sites to prevent influenza disease with a clear path towards clinical proof of correlation for protective efficacy in humans	(1) Biased viral samples may not generate consensus sequences that represent full profile of conserved immunogenicity; (2) Large efforts on surveillance data required	Pre-clinical tests on H1, H3 and H5 HA [28,41,42,43,45]
**Ancestral sequence reconstruction**	Figure 2B	Pre-clinical	(1) Induce broad cross-reactive protection within highly diverse influenza subtype (2) Account for sampling bias and the variability of substitution rates among sites; (3) Potentially avoid the detrimental effects of antigenic drift with ancestral sequences; (4) Incorporate protein functional and structural domains	(1) More sophisticated and advanced models to incorporate protein domains are still under development; (2) Experimental data on protein function is needed	Pre-clinical tests on ancestral sequence of H5N1 HA and NA [44]
**Immunomics**	Figure 2C	Pre-clinical & Clinical	(1) Account for the heterogeneity of the major histocompatibility complex (MHC) in host; (2) Protections and viral clearance from T-cell response has been distinctively tested	(1) Indirect estimation on epitope affinity to MHC; (2) To keep conformational epitopes to be function when designed into vaccine can be challenge	FP-01.1 Flu-v Multimeric-001 See Table 2 for details

**Table 2 vaccines-07-00045-t002:** Promising epitope-based universal influenza vaccines at clinical trials.

Vaccine	Company	Projects	Clinical Phase			Clinical Trial Registration#	Reference
			I	II	III		
**FP-01.1**	Immune Targeting Systems Ltd., London, UK.	FP-01.1	completed	completed		NCT01265914, NCT01677676, NCT02071329	Francis 2015 [80]
		FP-01.1-Adjuvant	completed			NCT01677676	unpublished
		FP-01.1 + seasonal TIV + FP-01.1-Adjuvant	completed			NCT01701752	unpublished
**Flu-v**	PepTcell Limited	Flu-v	completed			NCT01226758, NCT01181336	Pleguezuelos 2015 [82]
		adjuvanted Flu-v		completed		NCT03180801, NCT02962908	van Doorn 2017 [83]
**Multimeric-001 (M-001)**	BiondVax Pharmaceuticals Ltd	M-001	completed	completed		NCT01146119, NCT01010737	Atsmon 2014 [86]
		M-001 (prime) + seasonal TIV vaccine (boost)	completed	completed		NCT03058692, NCT01419925, NCT02293317	Atsmon 2014 [86]
		M-001 (prime) + H5N1 vaccine (boost)	completed	completed		NCT02691130	unpublished
		M-001 as standalone vaccine			ongoing	NCT03450915	unpublished

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
