# Peer review of "Computational Approaches and Challenges to Developing Universal Influenza Vaccines"

_vaccines, 2019, doi:10.3390/vaccines7020045_

Round 1

Reviewer 1 Report

The review by Qiu et al. provides a cogent summary of the application of computational approaches in the design of influenza vaccines for broad and durable protection.  The authors highlight the depth of data being accumulated by studies into viral diversity and host immunity, that may be better harnessed by improved computational analyses to optimize vaccine design.  The review is generally well written and of high interest to the field of universal influenza vaccine design.

Major comments:

1 - The review lacks tangible examples of where computational design have actually resulted in superior vaccines or vaccine strain selection than traditional approaches. While the premise is probably correct that improved computation of existing and future data may eventually lead to better vaccine designs. The review would be strengthened by highlighting many more illustrative publications where computational approaches actually drives favourable immunisation outcomes.

Minor comments:

Line 98 - durable immunity for all seasonal and even emerging pre-pandemic strains

Line103 – “While global surveillance efforts and data sharing agreements have increased available information, vaccine design often ignores the underlying processes of the global influenza meta-population which generates diversity that allows the viral populations to escape, immune, vaccine and anti-viral selection pressure.”  This sentence needs rewriting to make sensical.

Line 106 – “Furthermore, hemagglutination inhibition assays, central to vaccine strain selection, is a poor approximation for the average immune response that does not account for heterogeneity in the human immunogenetics due to diverse Major Histocompatibility Complex (MHC) genes [24]”.  While HAI assays have many limitations, it is unclear why this serological readout would be majorly influenced by host MHC haplotypes. Maybe clarify this concept.

Line 176 – “immunodominance” - I think you mean immunogenicity

Line 186 – “…identify potential vaccine antigens is called…”

Line 249 – “participants aged above 50 year for two years” – Unclear.

Line 258 – former?

Line 277 – in instead of “into”

Line 277 –“ The library then can be used into computational approaches to quantitatively measure the immune responses and further to predict the effects of vaccine candidates without costly animal tests.”

It is unclear how the wholesale sequencing of pathogen specific BCRs/antibodies can actually be translated into predications of vaccine candidates.  Presumably this required better models of antibody/epitope interaction in the absence of epitope localisation information. The authors should expand this section to make clear the current limitations with deriving an antibody epitope from sequence alone and whether this is actually realistic.

Line 315 – replace “under” with within

Line 318 – “With limited data, the consensus sequence may not effectively identify the conserved information along viral evolving history.” Unclear what the authors mean by this. Suggest clarification.

Line 319 “Furthermore, since antigenic drift is so rapidly occurring in influenza viruses, accurate prediction for future strains is barely possible”. 

This sentence is unnecessarily dramatic. All future strain prediction might be rendered impossible by drift. However, current WHO strain selection still gets it right at least some of the time, illustrating that even using crude ferret sera metrics, some predictive value is there. Also, not all mutations (in HA) are equally tolerated, with mutations heavily constrained by the necessity to result in a virus that is replicative and transmissible. This severely limits the evolutionary pathways available to virus and constrains available drift locations and residue choices, hence the focus on consensus or ancestral strains for vaccine design.

Line340 – “enhance the cross-reactivity and availability,”  sentence trails off and doesn’t really make sense.

Line379 – circulating potential

Line 376 – Section “Pathogen Evolvability” – this paragraph is very repetitive, making the same point repeatedly. Suggest rewriting for clarity.

Line408 – “Computational design highly relies on the data quality, including the representative of the samples, the completion and precision of data information, and other supplemental data for specific analysis purpose, for example, experimental testing results.”  This sentence is unclear and needs to be rewritten for clarity.

Line 442 – “Furthermore, model selection procedure should be applied during the process of model development to optimize the balance of biological realistic and computing power”    This sentence is unclear and needs to be rewritten for clarity.

Author Response

Reviewer 1: Comments and Suggestions for Authors

The review by Qiu et al. provides a cogent summary of the application of computational approaches in the design of influenza vaccines for broad and durable protection.  The authors highlight the depth of data being accumulated by studies into viral diversity and host immunity, that may be better harnessed by improved computational analyses to optimize vaccine design.  The review is generally well written and of high interest to the field of universal influenza vaccine design.

Thank you for your valuable and detailed comments. All revisions have been made for each of the comment as listed below. 

Major comments:

1 - The review lacks tangible examples of where computational design have actually resulted in superior vaccines or vaccine strain selection than traditional approaches. While the premise is probably correct that improved computation of existing and future data may eventually lead to better vaccine designs. The review would be strengthened by highlighting many more illustrative publications where computational approaches actually drives favourable immunisation outcomes.

To highlight each approach, we have added Table 1 that includes advantages and disadvantages of each approach. Table 1 and Table 2 together give a brief summary of preclinical and clinical examples. Table 1 is new addition to the review manuscript (shown below). 

Minor comments:
Line 98 - durable immunity for all seasonal and even emerging pre-pandemic strains

Done

Line103 – “While global surveillance efforts and data sharing agreements have increased available information, vaccine design often ignores the underlying processes of the global influenza meta-population which generates diversity that allows the viral populations to escape, immune, vaccine and anti-viral selection pressure.”  This sentence needs rewriting to make sensical.

Thank you. It has been rewritten as follows (Lines 131-134)

While global surveillance efforts and data sharing agreements have increased available information, vaccine design often ignores the underlying processes of the global influenza meta-population which generates diversity that allows the viral populations to escape vaccine-induced immune responses and anti-viral treatments”.

Line 106 – “Furthermore, hemagglutination inhibition assays, central to vaccine strain selection, is a poor approximation for the average immune response that does not account for heterogeneity in the human immunogenetics due to diverse Major Histocompatibility Complex (MHC) genes [24]”.  While HAI assays have many limitations, it is unclear why this serological readout would be majorly influenced by host MHC haplotypes. Maybe clarify this concept.

Here we tried to emphasize that the standard criteria HAI assay used for vaccine selection does not account for all processes and interactions between hosts and pathogens. The sentence has been revised as follows (Lines 135-138)

Furthermore, hemagglutination inhibition assay, central to vaccine strain selection, is a poor approximation for the average immune response that does not account for the heterogeneity of immune responses between hosts and pathogens, which cannot provide a full profile of pathogen immunogenic features [33].”

Line 176 – “immunodominance” - I think you mean immunogenicity

Done

Line 186 – “…identify potential vaccine antigens is called…”

Done

Line 249 – “participants aged above 50 year for two years” – Unclear.

We have revised it as follows “participants with age of older than 50 for a two-year follow-up.”

Line 258 – former?

Done

Line 277 – in instead of “into”

Done

Line 277 –“ The library then can be used into computational approaches to quantitatively measure the immune responses and further to predict the effects of vaccine candidates without costly animal tests.”

It is unclear how the wholesale sequencing of pathogen specific BCRs/antibodies can actually be translated into predications of vaccine candidates.  Presumably this required better models of antibody/epitope interaction in the absence of epitope localisation information. The authors should expand this section to make clear the current limitations with deriving an antibody epitope from sequence alone and whether this is actually realistic.

We agree that deriving an epitope from sequence alone is not as simple or straight forward as it sounds. Application of NGS to construct the antibody lineages with HIV genetic data have been introduced in this review by He et al [30]

We have discussed the current limitations with deriving an antibody epitope from sequence alone and added the section below (Lines 332-341).

“The main limitation with this approach is that linear sequence may not accurately predict the conformational variations when these antigens are put back in a complete protein context [94]. When the conformational structure of the epitope is not accurate, the corresponding immune response cannot be precisely computed [95]. To solve this issue, some high-performance bioinformatics tools such as molecular dynamics simulations can be used to predict the 3-D structure and stability of proteins or peptides [96], [97]. Furthermore, in the previous section, the successful maintenance of the conformational epitope in these clinical tested vaccines has provided positive evidence for epitope-based universal vaccine design. Taken together, with this antibody repertoire analysis tool, the computational estimation of immune stimulation of these predicted viral antigens in hosts can be more accurate. “

Line 315 – replace “under” with within

Done

Line 318 – “With limited data, the consensus sequence may not effectively identify the conserved information along viral evolving history.” Unclear what the authors mean by this. Suggest clarification.

The quality of a consensus sequence is more impacted by sampling bias than other approaches. Ideally, all existing pathogens are collected and used for generating a full representative consensus sequence. But since the surveillance epidemiological and genetic data from epidemics are only a small sampling of the population, the consensus sequence from these samples by COBRA approach may not present the full profile. The sentence with previous one has been revised to clarify (lines 376 – 382 in the revised version).

However, there are some major concerns with this approach. To be universally cross-reactive, the ideal CORBA HA protein is to cover all the conserved information within one subtype or multiple subtypes. The conserved immunogenic profile of consensus sequences from COBRA approach is dependent upon the sharing of epidemiological and genetic data collected during public health investigations and surveillance of outbreaks. With biased viral samples, the consensus sequence generated may not represent the full profile of conserved immunogenicity along viral evolving history.

Line 319 “Furthermore, since antigenic drift is so rapidly occurring in influenza viruses, accurate prediction for future strains is barely possible”.  
This sentence is unnecessarily dramatic. All future strain prediction might be rendered impossible by drift. However, current WHO strain selection still gets it right at least some of the time, illustrating that even using crude ferret sera metrics, some predictive value is there. Also, not all mutations (in HA) are equally tolerated, with mutations heavily constrained by the necessity to result in a virus that is replicative and transmissible. This severely limits the evolutionary pathways available to virus and constrains available drift locations and residue choices, hence the focus on consensus or ancestral strains for vaccine design.

This has been modified as follows (Lines 382-386):

“Even with increased global efforts to collect data and characterize epidemics it is unlikely that sufficient data could be collected to overcome this challenge. Alternative approaches, such as phylogenetic modeling of viral proteins along a characterized evolutionary trajectory that account for impacts of sample biases and missing data could greatly improve design of COBRA candidates.”

Line340 – “enhance the cross-reactivity and availability,”  sentence trails off and doesn’t really make sense.

We have changed the sentence to “enhance the cross-reactivity” in line 409. 

Line379 – circulating potential

Done

Line 376 – Section “Pathogen Evolvability” – this paragraph is very repetitive, making the same point repeatedly. Suggest rewriting for clarity.

We have revised this section to reduce repeated points in lines 449-475. 

“Uncovering the important ecological, immunological and environmental determinants on viral evolution is very important to make predictions of the viral emergence, fitness, transmissions and circulating potential after new substitution is introduced [123]. Evolvability, first coined by Kirschner and Gerhart in 1998, means that the organism’s capacity to generate heritable phenotype [124]. The zoonotic nature and complicated ecology of influenza viruses make evolvability more difficult to quantify and predict. But with the advances of phylogenetic algorithms, models can integrate and evaluate the impacts of environmental determinants. For example, an important development in phylogenetic modeling was the field of viral phylodynamics that was introduced in 2004 to study “how epidemiological, immunological, and evolutionary processes act and potentially interact to shape viral phylogenies” [125], [126]. Dynamics of influenza virus infections and transmissions at individual-level (such as viral evolution within an infected host), population-level (individual hosts within a population), or ecology-level (entire populations of different host species) have been studied [125]. Specially, phylodynamics have been used to study factors of interest on some viral phenotypes, including virulence, viral transmissibility, cell or tissue tropism, and antigenic phenotypes that can facilitate immune escape, etc. [108], [125]. Details of methods and examined significant factors can be found in these reviews [108], [125], [126].

Furthermore, the complements between phylodynamic modeling and experimental testing can be integrated together to improve prediction on influenza virus evolvability. For example, experimental studies designed to assess viral evolvability [127], [128]demonstrated that a measured fitness score or estimated tolerance for mutations can be used in phylodynamic modeling to link phenotypes, genetic characteristics and other ecological factors, which can improve the prediction of viral evolvability for natural influenza virus strains [128]. The potential predictors and consequent mutations computed by models can enhance our understanding on viral characteristics, potential immune escape, or influenza antiviral drug resistance [129]. Challenges for this area are how to get accurate and sufficient information on the epidemiological, immunological and ecological factors, how to expand, integrate and enhance phylodynamic models [108], and how to gather the current modeling factors to improve prediction of viral evolvability [123].”

Line408 – “Computational design highly relies on the data quality, including the representative of the samples, the completion and precision of data information, and other supplemental data for specific analysis purpose, for example, experimental testing results.”  This sentence is unclear and needs to be rewritten for clarity.

We have revised this sentence as below and lines 482-485 in the manuscript. 

“Computational vaccine design relies on the input data quality [30]. To be specific, the representative of the collected samples, the completion and precision of recorded data, and the timely manner of data sharing and availability can ameliorate the output from computational modeling [19], [130], [131].”

Line 442 – “Furthermore, model selection procedure should be applied during the process of model development to optimize the balance of biological realistic and computing power” This sentence is unclear and needs to be rewritten for clarity.

We have revised this sentence as follows (Lines 526-528).

“Furthermore, to avoid overparameterization, model selection procedure should be applied during the process of novel model development to optimize the balance of biological reality and parameterization [138], [141].”

Reviewer 2 Report

Qiu and colleagues review computational approaches for the design of a universal influenza vaccine and discuss their challenges and needs. The authors begin by describing the current vaccine selection process by the WHO and the drawbacks of vaccinating against seasonal influenza viruses. The authors summarize current methods and categorize them coarsely into model-free consensus-based approaches, phylogenetic model-based approaches and immunomics.  In the end, they finish with outlining requirements and an ideal set up for computational vaccine design.

General comments.

Consider revising the headlines of the following section:

Change section 1) from “current approach of influenza vaccine design and its advantage” to “current approach of seasonal influenza vaccine design and its advantage”. Change subsection 1.1) from “wild strain selection approach” to “selection of circulating influenza viruses”. In the following section, you only describe the WHO selection process.

Change subsection 2.2.3.) from “Computational approaches to incorporate host immunological factors” to “Computational approaches that incorporate host immunological factors”.

Missing references; please add a reference at the end of the following sentences: line 53, line 112, line 151, line 363, line 387, line 430

- cited references do not seem appropriate in several places, please check - and see below for some examples (not a complete list)

- Please describe in section 1 current approaches to universal vaccines. You can also refer to these two reviews about computational vaccine prediction of seasonal viruses (10.1016/j.tim.2017.09.001 and doi: 10.1016/j.tim.2017.09.004)

Line 45: Consider rephrasing the sentence, it sounds like wet-lab experiments benefit from computational techniques.

Line 63: What is a wild strain? Do you mean wild type? Please define.

Line 69. Strains are selected twice a year, not every two years.

Line 74: Vaccine production for seasonal influenza vaccine last 9-12 months. Please see, doi: 10.1016/j.tim.2017.09.001 and doi: 10.1016/j.tim.2017.09.004

Line 88: Use “significant” only when you can provide a p-value, please rephrase.

Line 102f.: What kind of pathogens do you mean? Influenza viruses or others?

Line 102: What do you mean by stochastic nature? Please define.

Line 105f.: What do you mean by an immune vaccine? And how does it related to antiviral immune pressure. Please, elaborate more on that.

Line 107: MHC genes have nothing to do with HI assays that measure B-cell responses.

Line 133f.: In line 103f. you describe that data sharing has increased without strong effects on influenza vaccine design, you now declare that the increasing amount of data is advantageous for vaccine design.

-adding citations for the commonly used databases in this area would be appropriate here.

Line 142: What do you mean by “computing protein structure prediction”? Do you mean predicting protein structure or computing protein structure?

Line 143f.: “4) machine learning to incorporate viral, ecological, epidemiological and host immunological data to make precise assessment and prediction [26]: sounds intriguing, but unfortunately ref 26 has nothing do do with machine learning, and seems to focus on parasites, not viruses? It is behind a pay-wall, so going by the abstract and title.

Line 162f.: Please describe ML and Bayesian inference methods and add a reference regarding that they are “most commonly used”.

Line 164f.: Do not understand this sentence. Do you mean: “evolutionary models that incorporate …. “?

Line 166f.: Please clarify. What do you mean here? Do you mean inference of head vs stalk domain within the groups or within the subtypes or between them?

Line 170. Please describe how these epitopes are computed.

Line 175: How are “whole packaged virions” produced? This is not described before.

Line 180: Please remove “professional” from this sentence. Otherwise, specify what you want to say.

Line 209f.: How do HMM, SVM, ANN, hybrid, consensus and pan-specific methods work. Please describe these techniques and add references.

Line  228 and 232: Do you mean T-cell epitopes? Otherwise please also describe B-cell approaches.

Line 237: Don’t understand the number. Is each peptide 35 amino acids long?

Line 265f.: Please describe how antibody affinity can be analyzed.

Line 288f.: A consensus sequence does not capture conserved epitopes. The complete sequence is conserved. What do you want to say here? Or do mean a profile?

Line 292f.: How can a VLP maintain conformational epitopes? Please clarify.

Line 298: What is a geo-location? Do you mean a geographical location?

Line 301: A consensus sequence does not reflect genetic diversity. It is conserved at each position.

Line 313. How does COBRA design strains with HA head domains? Please explain how COBRA works.

Line 316: What is the quality of a consensus sequence? What do you mean here? How does it rely on surveillance data? What is surveillance data? Please clarify.

Line 319: Please explain, why and how antigenic drift effect accurate vaccine prediction.

Line 324: How can ancestral character state reconstruction reveal “some conserved properties”? What do you mean by “some conserved properties”? Ancestral character state reconstruction infers hypothetical precursor sequences of sampled viruses.

Line 328: How are phylogenetic models used to reconstruct a common ancestor? Please describe. Please do not get confused with models and algorithms.

Line 339: What is a seed strain?

Line 341f.: What is a clade-specific computational derived ancestral sequence? Please explain this in detail? How is this method working?

Line 345: How does the structure of the protein influence the inferred phylogeny? Please explain in detail.

Line 352: Please, start the sentence with an article: “The integration …”

Line 354: What method assumes site independence? Where is site independence coming from? Why does it not capture biological reality?

Line 355: How do some modeling methods overcome these challenges? Which are able to do that? Why is it a problem, when methods exist that can handle these issues?

Line 362: Please, name these methods? How does a more detailed representation or proteins look like? Please explain.

Line 366+379: Replace “mutation” with “substitution”.

Line 372: Remove “most nucleotide” form sentence.

Line 379: What do you mean with “potential circulating”? Please explain.

Line 390: I do not understand what you are trying to say here: “With these models inferential models”?

Line 391: What is the population and what is the ecology level? Please explain

Line 391: The sentence starting with “These factors from …” is hard to understand and a bit too long. Consider breaking it into three sentences.

Line 397: What are key predictors? What do you mean by “identified critical mutations”? How can they facilitate our understanding? Consider using a different word than “facilitate”. Explain in detail.

Line 409: What is “other supplemental data”? Please explain.

Line 414f.: Please consider adding that timely availability of sequence data is a third missing issue. Please see, doi: 10.1016/j.tim.2017.09.001 and doi: 10.1038/s41598-017-18791-z

Line 416f.: How should such policies look like? Please describe/explain.

Line 417f.: Why is the imbalance sampling for zoonotic influenza more severe? Please explain.

Line 419: Replace “zoonosis” with “zoonotic”. Or what are you trying to say here?

Line 420: The sentence “which are recognized as the natural reservoir of avian influenza A viruses (AIV) [111]” should go to the introduction.

Line 425f.: Please explain why “a sample-based accumulation curve” can help in identifying virus subtypes.

Line 428: Why is there an issue with data records, please explain

Line 433: Why and how would such a protocol help? Please explain.

Line 436: I do not see how and where in the previous section your statement has been shown/explained.

Line 437: How do we benefit from more advances in computer power? Please explain.

Line 438f.: What would be the effect of such a viral phylodynamic model?

Line 443: Why is this the ultimate goal? Machine learning is barely mentioned before. Please explain in detail how it can help.

Line 462: Definition of genetic drift is missing. Same for antigenic drift, mentioned before. However antigenic shift is not mentioned at all.

Author Response

Reviewer 2:

Comments and Suggestions for Authors
Qiu and colleagues review computational approaches for the design of a universal influenza vaccine and discuss their challenges and needs. The authors begin by describing the current vaccine selection process by the WHO and the drawbacks of vaccinating against seasonal influenza viruses. The authors summarize current methods and categorize them coarsely into model-free consensus-based approaches, phylogenetic model-based approaches and immunomics.  In the end, they finish with outlining requirements and an ideal set up for computational vaccine design.

General comments.

Consider revising the headlines of the following section:
Change section 1) from “current approach of influenza vaccine design and its advantage” to “current approach of seasonal influenza vaccine design and its advantage”. Change subsection 1.1) from “wild strain selection approach” to “selection of circulating influenza viruses”. In the following section, you only describe the WHO selection process.
Change subsection 2.2.3.) from “Computational approaches to incorporate host immunological factors” to “Computational approaches that incorporate host immunological factors”.

Thank you for the comments. We have made the changes for the headlines as suggested.

Missing references; please add a reference at the end of the following sentences: line 53, line 112, line 151, line 363, line 387, line 430

Reference have been added and listed below.

Original line 53, now line 52:

However, it has found limited applications in the design and discovery of new vaccines, an area where proper integration of computational support and design is essentially needed [2], [6].

Original line 112, now line 145:

As a consequence, traditional design inefficiently captures few pathogen features based on a limited input that does not account for the high diversity of pathogen and high heterogeneity of host’s immune responses [2][34].

Original Line 151, now line 169:

Computational approaches with advance data integration and quantitative empirical analyses fit the needs of universal vaccine design for highly diverse influenza virus in several promising aspects [38], [39]: 1) …

Original Line 363, now line 420:

Precise estimation of influenza virus evolution including protein structural and its functional information supported by experimental data [112], may help to efficiently identify and select target antigens for universal vaccine design [30] 

Original Line 387, now line 446:

This information of protein function from rapid high-throughput experiments may greatly improve the precision of ancestral sequence reconstruction [122].

Original Line 430, now line 492:

The unequal sampling of geographical regions is caused by global and local resource allocation [133].

- cited references do not seem appropriate in several places, please check - and see below for some examples (not a complete list)

Thank you for the comments. We have rechecked all citations and made sure they are proper and have been inserted at the intended places in the context. 

- Please describe in section 1 current approaches to universal vaccines. You can also refer to these two reviews about computational vaccine prediction of seasonal viruses (10.1016/j.tim.2017.09.001 and doi: 10.1016/j.tim.2017.09.004)

Thank you for bringing this up. We have summarized the predictive models and added a new section to summarize current approaches to universal vaccine design as below and lines 94-126 in the revised version.

“Predictive models of viral evolution to forecast dominant circulating influenza viral strains in the upcoming influenza seasons through the analysis of genetic and epidemiological data from influenza surveillance system have been developed to make quantitative predictions of viral evolution and aim to improve the selection of seasonal influenza vaccine candidates [10], [19]. This framework has demonstrated potential to integrate multiple data sources to improve influenza vaccine design.

1.2 Universal influenza vaccine design

The seasonal vaccines offer a little or no protection to emerging zoonotic influenza viruses with pandemic potential, as many species, especially wild aquatic birds, are recognized as the natural reservoir of all subtypes of influenza A viruses and have the potential to occur spillover and infect humans directly [20]. As with past pandemics, the surface glycoproteins, HA and neuraminidase (NA) are replaced through reassortments of zoonotic strains where the human population has no pre-existing immune protection and the vaccines in use are not cross-reactive with these new strains [21]–[23]. Experimentally identified conserved and immunogenic M2 protein antigens [24], and HA-stalk design [24]–[26]have potential to elicit broadly protective antibodies against seasonal influenza strain. M2-based universal vaccine design focuses on the conserved antigens that have been experimentally identified on M2 protein. However, the low immunogenicity and epitope density by viral nature has been a fatal limit to make the cross-protection from M2 being effectively applied into vaccine design [24]. To solve this issue, many approaches have been developed to improve M2 immunogenicity, details of which can be found in this review by Zhang et al [24]. Similar with M2-based design, HA-stalk design tries to elicit the conserved and cross-reactive protection from the membrane-proximal stalk domain [25]. While the stalk domain is conserved across multiple influenza subtypes, it is shielded by the immune-dominant head domain. To amplify the broad protection from stalk domain, truncated HA without head domain, concentrated short peptides from stalk domain or recombinant chimeric HA proteins have been employed [24]–[26]. Despite the potential for both M2 and HA-stalk design vaccines to elicit broadly reactive immune response, a number of challenges remain (reviewed in [24]and [27]), including a limited understanding of the full repertoire of potential epitopes. More systematic computational approaches that go beyond circulating strain prediction and incorporate a full profile of antigens stimulating both humoral and cellular immune responses are needed for universal vaccine design [24]–[26]. To overcome these challenges, computational approaches have been employed to rationally and promisingly design vaccine candidates that can induce broadly (ideally universally) cross-protective and durable immunity for all seasonal and even emerging pre-pandemic strains [13], [28], [29].“

Line 45: Consider rephrasing the sentence, it sounds like wet-lab experiments benefit from computational techniques.

This sentence has been revised as below at lines 45-47. 

Recent approaches to vaccine design has taken advantage of large-scale viral sequencing platforms, phylogenetic frameworks, protein structural modeling and systems biology to design novel broadly-reactive vaccine candidates, which have been used for influenza and other pathogens [2].”

Line 63: What is a wild strain? Do you mean wild type? Please define.

Wild strain means “natural influenza virus strains”. To avoid confusion, we have revised the headline to “Selection of circulating influenza viruses for seasonal vaccine design”.

Line 69. Strains are selected twice a year, not every two years.

We have corrected it to “twice a year”. 

Line 74: Vaccine production for seasonal influenza vaccine last 9-12 months. Please see, doi: 10.1016/j.tim.2017.09.001 and doi: 10.1016/j.tim.2017.09.004

Thank you for the details. We have checked the papers as you recommended. It (doi: 10.1016/j.tim.2017.09.004) says,

“Because it takes at least 6–8 months to develop and produce an updated influenza vaccine, scientists must decide which influenza virus variants to include in the vaccine nearly a year in advance.” 

And 

“Emerging influenza variants with high probabilities of spread in upcoming seasons must therefore be identified as fit and developed into CVVs at least 9–12 months in advance of their possible inclusion in the vaccine.”

With this information, what we tried to indicate here is that it takes a long period of time to produce the vaccine after selection. We believe that “at least 6-8 months” is correct in this context. 

Line 88: Use “significant” only when you can provide a p-value, please rephrase.

We have rephrased “significant” to “relevant functional” at line 90 in the revised version.

Line 102f.: What kind of pathogens do you mean? Influenza viruses or others?

Here we talked in a broader sense of hypervariable and rapidly-evolving viral pathogens. To avoid confusion, we have added, “including influenza viruses” at line 130 in the revised version.

Line 102: What do you mean by stochastic nature? Please define.

We have clarified this sentence as follows (Lines 130-131).

Reasons for failure include inherent uncertainty in pathogen evolution [32].”

Line 105f.: What do you mean by an immune vaccine? And how does it related to antiviral immune pressure. Please, elaborate more on that.

We have revised this as follows (Lines 131-134).

While global surveillance efforts and data sharing agreements have increased available information, vaccine design often ignores the underlying processes of the global influenza meta-population which generates diversity that allows the viral populations to escape vaccine-induced immune responses and anti-viral treatments.”

Line 107: MHC genes have nothing to do with HI assays that measure B-cell responses.

Here we tried to emphasize that the standard criteria HAI assay used for vaccine selection only captures one measurement of a complex system that determines genetic and antigenic diversity of circulating influenza viruses. 

The sentence has been revised as follows (Lines 136-139).

“Furthermore, hemagglutination inhibition assay, central to vaccine strain selection, is a poor approximation for the average immune response that does not account for the heterogeneity of immune responses between hosts and pathogens, which cannot provide a full profile of pathogen immunogenic features [33].”

Line 133f.: At line 103f. you describe that data sharing has increased without strong effects on influenza vaccine design, you now declare that the increasing amount of data is advantageous for vaccine design. 

-adding citations for the commonly used databases in this area would be appropriate here.

The commonly used databases have been added, including the Global Initiative on Sharing All Influenza Data (GISAIDhttps://www.gisaid.org/), the FluID epidemic data (https://www.who.int/influenza/surveillance_monitoring/fluid/en/) and the NCBI Influenza Virus Database (https://www.ncbi.nlm.nih.gov/genomes/FLU/Database/nph-select.cgi?go=database). We have added these citations.

Line 142: What do you mean by “computing protein structure prediction”? Do you mean predicting protein structure or computing protein structure?

Thank you for capturing the ambiguous description. Here we talk about potentials of computational design. One way is to incorporate protein functional domains into the model to simulate the selection pressure on these domains. Therefore, we have revised the sentence to “capability of incorporating protein functional structure and antibody repertoire analysis via structural biology.” at lines 172-173 in the revised version.

Line 143f.: “4) machine learning to incorporate viral, ecological, epidemiological and host immunological data to make precise assessment and prediction [26]: sounds intriguing, but unfortunately ref 26 has nothing do do with machine learning, and seems to focus on parasites, not viruses? It is behind a pay-wall, so going by the abstract and title. 

Sorry for the confusion. We put the reference at an improper location. We have moved this reference to the correct location at line 169.

Computational approaches with advance data integration and quantitative empirical analyses fit the needs of universal vaccine design for highly diverse influenza virus in several promising aspects [38], [39]: 1)…

Line 162f.: Please describe ML and Bayesian inference methods and add a reference regarding that they are “most commonly used”.

We have added reference and edited these sentences at lines 193-196. We are unsure how to add this section and treat it fairly in a figure legend. We have added references hoping this helps

“Maximum likelihood and Bayesian approaches are the most commonly used statistical phylogenetic methods to reconstruct ancestral sequence at the ancestral node (shown as black dot on the tree) [45]–[47].”

Line 164f.: Do not understand this sentence. Do you mean: “evolutionary models that incorporate …. “?

This has been revised as suggested

Line 166f.: Please clarify. What do you mean here? Do you mean inference of head vs stalk domain within the groups or within the subtypes or between them?

Thank you for the question. Corresponding with the Fig 2B, the ancestral sequence can be reconstructed for head domain and stalk domain independently on the same phylogeny. This can be conducted within the groups: one set for Group 1 and one set for Group 2 influenza A viruses. To clarify, we have revised the sentence at lines 198-202 in the revised version.

“Based on the evolutionary relationship among different subtypes of influenza A virus, common ancestral sequences of head and stalk domains can be generated within influenza A virus Group 1 (H1, H2, H5, H6, H7, H8, H9, H11, H12, H13, H16, H17, and H18) and within Group 2 (H3, H4, H7, H10, H14, and H15), respectively.”

Line 170. Please describe how these epitopes are computed.

T-cell epitope prediction tools have been used to compute these epitopes. Motif-search, a type of sequence-based method, uses information from the anchor residues of peptide sequences known to bind to MHC to predict the peptide-MHC binding affinity. Subsequently the composition and length of non-anchor residues were shown to influence the ability of peptide to bind to MHC, which led to the development of matrix motif (MM) methods. Based on MM methods, quantitative matrix (QM) methods were developed that account for known information on peptides and their corresponding binding affinities to predict affinity. Since traditional sequence-based methods rely on experimental data, predictions for less prevalent MHC alleles was limited by lack of training data. Hence, machine learning approaches like artificial neural networks (ANN), support vector machines (SVM), HMM (hidden markov models) were developed that extend known peptide-MHC binding affinity data to uncharacterized MHC alleles. The consensus methods (combinations of two or more different methods) exhibited higher prediction accuracy than the use of single machine learning methods. Structure-based methods use the information from the biochemical interaction of peptide and MHC, to make predictions. EpiDOCK, is the first structure-based server to predict peptide-MHC class II binding affinity and reported an overall accuracy of 83% with 90% of sensitivity and 76% of specificity.

We have added some references in the main context to cover the computation of epitopes in lines 245 -247 in the revised version.

“2) selection of epitope prediction tools may also introduce discrepancy in the identification of potential T-cell epitopes due to methodological differences. T-cell epitope prediction tools that include sequence- and structure-based methods are reviewed in Patronov et al [55]an Luo et al [62]);”

Line 175: How are “whole packaged virions” produced? This is not described before.

We have replaced “whole packaged virions” with “virus-like particles (VLPs)” in this section. The sentence has been revised as follows (Lines 209-210). 

“All outputs from these three approaches, like epitopes, peptides, proteins or virus-like particles (VLPs), will be tested at in-vitro and/or in-vivo models to evaluate their immunogenicity.”

Line 180: Please remove “professional” from this sentence. Otherwise, specify what you want to say.

Done

Line 209f.: How do HMM, SVM, ANN, hybrid, consensus and pan-specific methods work. Please describe these techniques and add references.

These approaches are described as following. Motif-search, a type of sequence-based method, uses information from the anchor residues of peptide sequences known to bind to MHC to predict the peptide-MHC binding affinity. Subsequently the composition and length of non-anchor residues were shown to influence the ability of peptide to bind to MHC, which led to the development of matrix motif (MM) methods. Based on MM methods, quantitative matrix (QM) methods were developed that account for known information on peptides and their corresponding binding affinities to predict affinity. Since traditional sequence-based methods rely on experimental data, predictions for less prevalent MHC alleles was limited by lack of training data. Hence, complex machine learning approaches like artificial neural networks (ANN), support vector machines (SVM), HMM (hidden markov models) were developed that extend known peptide-MHC binding affinity data to uncharacterized MHC alleles. The consensus methods (combinations of two or more different methods) exhibited high prediction accuracy than the use of single machine learning methods. Structure-based methods use the information from the biochemical interaction of peptide and MHC, to make predictions. EpiDOCK, is the first structure-based server to predict peptide-MHC class II binding affinity and reported an overall accuracy of 83% with 90% of sensitivity and 76% of specificity.

To avoid the distractions from these details, we have removed the names of these approaches and added references of related reviews in the main context to cover the computation of epitopes in lines 245 -247 in the revised version.

“2) selection of epitope prediction tools may also introduce discrepancy in the identification of potential T-cell epitopes due to methodological differences. T-cell epitope prediction tools that include sequence- and structure-based methods are reviewed in Patronov et al [55]an Luo et al [62]);”

Line  228 and 232: Do you mean T-cell epitopes? Otherwise please also describe B-cell approaches.

Thank you. Yes, given the reliability in the T-cell epitope predictions, and evidence of epitope-based vaccines at clinical stages (Table 2), we confined to the T-cell epitopes only. The text has been revised

Line 237: Don’t understand the number. Is each peptide 35 amino acids long?

We modified the sentence to “FP-01.1 vaccine (also called as Flunisyn™), comprises six different synthetic peptides (length: 35 amino acids) each conjugated to the fluorocarbon moiety C8F17(CH2)2-COOH” in lines 289-290 in the revised version.

Line 265f.: Please describe how antibody affinity can be analyzed. 

We have removed this sentence. The paragraph focuses on the analysis of antibody repertoire to characterize broadly neutralizing antibodies. This is discussed and reviewed in detail in the given reference [30]. It is based on the structural biology to evaluate the strength of antibody binding to the antigen. 

 [30]L. He and J. Zhu, “Computational tools for epitope vaccine design and evaluation.,” Curr. Opin. Virol., vol. 11, pp. 103–12, Apr. 2015

Line 288f.: A consensus sequence does not capture conserved epitopes. The complete sequence is conserved. What do you want to say here? Or do mean a profile?

It means that a consensus sequence can generate the linear profile of conserved genetic information, which eventually aims for capture some conserved epitopes on this genetic information. To be clear, the sentence has been changed as below and lines 345-347 in the revised manuscript. 

These sequences are expected to effectively capture a profile of conserved genetic and epitope information which can induce cross-reactive cellular immune responses [98].”

Line 292f.: How can a VLP maintain conformational epitopes? Please clarify.

The VLPs are similar to intact virions, which mostly keeps the three-dimensional structure of antigens. Before the antigen on the VLP is digested into linear epitopes, a receptor of the immune system can interact with its originally conformational structure. Because some epitopes may contain discontinuous amino acids in its three-dimensional conformation (this is called conformational epitopes), the VLP can maintain the 3-D conformation like the live virus but it is non-pathogenic.

Line 298: What is a geo-location? Do you mean a geographical location?

Yes. It has been fixed as “were sampled in different geographical locations”.

Line 301: A consensus sequence does not reflect genetic diversity. It is conserved at each position. 

We agree. To correct the expression, we have revised the sentence as below and lines 356-359 in the revised version.

If samples from one location or one host are overrepresented in the sequences used to generate consensus, then it can bias the output consensus sequence, which may not accurately represent the full conserved genetic profileof the whole H5N1 population.”

Line 313. How does COBRA design strains with HA head domains? Please explain how COBRA works. 

COBRA first generate HA consensus sequences. Then the COBRA HA structure is generated with 3D-JIGSAW algorithms. After transfected in human embryonic kidney, the COBRA HA protein is packaged as virus-like particles. Then functional characterization like receptor binding characteristics is examined, and antigenicity is determined by monoclonal antibody binding. For details, please refer to the paper (DOI: 10.1128/JVI.03152-15). Monoclonal antibodies could be against head domain or stalk domain or both. The reason we mentioned the HA head domain here is because COBRA claims that it has advantages over other universal vaccine candidates, for COBRA HA-elicited antibodies are able to against receptor binding site. To avoid confusion, we have revised the sentence as below and lines 371-372 in the revised version.

“COBRAs-based designed HA proteinof H1, H3 and H5 have been tested with in-vitro assays and animal models.”

Line 316: What is the quality of a consensus sequence? What do you mean here? How does it rely on surveillance data? What is surveillance data? Please clarify.

The representativeness for consensus sequence to describe the average immunogenicity profile of the viral population is dependent on which samples collected through routine surveillance are sequenced. Ideally, all existing pathogens are collected and used for generating a full representative consensus sequence. But since epidemiological surveillance and genetic data from epidemics only represent a small sampling of the population, the consensus sequence from these samples by COBRA approach may not present the full profile. The sentence has been revised to clarify in lines 376-382 in the revised version.

However, there are some concerns with this approach. To be universally cross-reactive, the ideal CORBA HA protein should contain all the conserved information present in multiple subtypes. The conserved immunogenic profile of consensus sequences from COBRA approach is dependent upon the sharing of epidemiological and genetic data collected during public health investigations and surveillance of outbreaks. With biased viral samples, the consensus sequence generated may not represent the full profile of conserved immunogenicity along viral evolving history. Even with increased global efforts to collect data and characterize epidemics it is unlikely that sufficient data could be collected to overcome this challenge. Alternative approaches, such as phylogenetic modeling of viral proteins along a characterized evolutionary trajectory that account for impacts of sample biases and missing data could greatly improve design of COBRA candidates.

Line 319: Please explain, why and how antigenic drift effect accurate vaccine prediction.

We have removed this comment. Please see response to reviewer 1 for more detail

Line 324: How can ancestral character state reconstruction reveal “some conserved properties”? What do you mean by “some conserved properties”? Ancestral character state reconstruction infers hypothetical precursor sequences of sampled viruses.

Thank you for the comment. We agree that ancestral sequence infers the precursor sequence of the sampled virus under the tested hypothetical phylogeny. But if the residues on the ancestral sequences are conserved in all the individuals within one lineage, presumably the phenotype and the potential cross reactivity of the ancestral virus would also be evolutionarily conserved. We have clarified this sentence as follows (Lines 400-402). 

Ancestral sequences can reveal conserved functions of the pathogen protein where the potential cross reactivity of the ancestral virus would also be evolutionarily conserved [104]. These conserved functions may indicate potential immune targets.”

Line 328: How are phylogenetic models used to reconstruct a common ancestor? Please describe. Please do not get confused with models and algorithms.

Thank you for bringing up the terminology issue. We have revised this part to also include how the common ancestor is reconstructed under a maximum-likelihood algorithm. 

Phylogenetic algorithms have been developed to reconstruct ancestral sequences for broadly-reactive vaccine design [44][107]. This phylogenetic approach with marginal reconstruction yields the maximum likelihood at the site with a specific amino acid after comparing all probabilities of different amino acids at a site on an internal node [107].”

Line 339: What is a seed strain?

We have revised it to “vaccine candidate sequences”.

Line 341f.: What is a clade-specific computational derived ancestral sequence? Please explain this in detail? How is this method working?

In this example with model-based computational approach, Ducatez et al, conducted analysis on H5N1,which contains diverse lineages defined by phylogenetic distances of HA genetic sequences as shown in the paper (doi: 10.1073/pnas.1012457108) Fig. 1. They constructed ancestral sequence for some lineages after their divergence. Based on a maximum likelihood tree, ancestral sequences were reconstructed at the internal nodes of co-circulating HPAI H5N1 viral lineages to capture the conserved genetic characteristics of these viruses. This phylogenetic approach with marginal reconstruction yields the maximum likelihood at the site with a specific amino acid after comparing all probabilities of different amino acids at a site on an internal node. 

To avoid confusion with a new term “clade”, we have removed “clade-specific” from the sentence. 

Line 345: How does the structure of the protein influence the inferred phylogeny? Please explain in detail.

Thank you for the comment. Protein functional or structural domains can be under different immunologic pressure during evolutionary process. For example, the head domain of HA is under higher selection pressure compared to stalk domain. To consider this biological reality, it may inaccurately infer one evolutionary rate of stalk and head domain together. We have revised the sentence as below and lines 414-415 in the revised version.

“The functional and structural domainsof pathogen protein can be under disparate immunologic pressures and thus have impacts on the evolutionary phylogeny [109]and the accuracy of ancestral sequence reconstruction.”

Line 352: Please, start the sentence with an article: “The integration …”

Done

Line 354: What method assumes site independence? Where is site independence coming from? Why does it not capture biological reality?

Currently, the nucleotide substitution models used in phylogenetic analysis are assuming nucleotide site independence, that is, the change of nucleotide on one site is independent from other sites. This does not capture biological reality, because some nucleotide sites may be linked together – one site change may result in higher probability of change in the linked sites (https://doi.org/10.1093/molbev/msy188). We have added the new reference at the end of the sentence at line 424. 

“2) The assumption of nucleotide site independence in the model cannot capture the biological reality that some sites are linked due to shared function [113].”

Line 355: How do some modeling methods overcome these challenges? Which are able to do that? Why is it a problem, when methods exist that can handle these issues?

We have two examples listed to explain these concepts. They both can deal with the two challenges listed above. Model selection has shown superiority of these models to handle the biological reality and accurate phylogeny reconstruction. To make it clear, we have revised it as below and at lines 425-431 in the revised version.

Some modeling approaches with protein structure scoring system or partitioning schemes on the protein sequence [97], [110], [114]can potentially overcome these challenge, for example, protein structure has been explored with coarse-grained models for structure prediction, prediction of protein interaction and molecular dynamics simulations of protein folding [97]. This provides the statistical potential like a scoring system for sequence-structure compatibility, which can be used to evaluate the probability of fixation of a given mutation and improve the precision of ancestral reconstruction [111].”

Line 362: Please, name these methods? How does a more detailed representation or proteins look like? Please explain.

We have presented some discussion on the current models in this revision. The more detailed representation of proteins is proposed in the next paragraph, which is called deep mutational scanning. We have revised them as below and lines 432-434 in the revised version.

Simple representations of protein functional and structural domains have been used so far. Hypothetically, novel models with a more complete representation with a full site mapping of the protein functions and structures would yield a better fit.

Line 366+379: Replace “mutation” with “substitution”. 

Done

Line 372: Remove “most nucleotide” form sentence.

Done

Line 379: What do you mean with “potential circulating”? Please explain. 

Revised it to “circulating potential” as suggested by reviewer 1. 

Line 390: I do not understand what you are trying to say here: “With these models inferential models”?

This paragraph has been revised. This phrase and its following sentence have been removed. 

Line 391: What is the population and what is the ecology level? Please explain

The population and ecology level have been defined in the sentence in lines 458-461 in the revised version.

“Dynamics of influenza virus infections and transmissions at individual-level (such as viral evolution within an infected host), population-level (individual hosts within a population), or ecology-level (entire populations of different host species) have been studied [125].”

Line 391: The sentence starting with “These factors from …” is hard to understand and a bit too long. Consider breaking it into three sentences.

This sentence repeats is repetitive and has been removed. 

Line 397: What are key predictors? What do you mean by “identified critical mutations”? How can they facilitate our understanding? Consider using a different word than “facilitate”. Explain in detail.

Apologies for the confusion. Here we tried to include what information related to viral evolvability can be generated with computational modeling. With integration of modeling and experimental tests, we could identify significant ecological, epidemiological and immunological factors (predictors) and important mutations to predict viral evolvability. Revisions have been made as follows (Lines 465-475).

“Furthermore, the complements between phylodynamic modeling and experimental testing can be integrated together to improve prediction on influenza virus evolvability. For example, experimental studies designed to assess viral evolvability [127], [128]demonstrated that a measured fitness score or estimated tolerance for mutations can be used in phylodynamic modeling to link phenotypes, genetic characteristics and other ecological factors, which can improve the prediction of viral evolvability for natural influenza virus strains [128]. The potential predictors and consequent mutations computed by models can enhance our understanding on viral characteristics, potential immune escape, or influenza antiviral drug resistance [129]. Challenges for this area are how to get accurate and sufficient information on the epidemiological, immunological and ecological factors, how to expand, integrate and enhance phylodynamic models [108], and how to gather the current modeling factors to improve prediction of viral evolvability [123].”

Line 409: What is “other supplemental data”? Please explain.

Other supplemental data was meant the experimental testing data if the strain has been tested, for example, the information of antigenicity and virulence. This literally could be a point of the completion of records. To incorporate comments below, we have revised this sentence at lines 482-485 in the revised version.

“Computational vaccine design highly relies on the input data quality [30]. To be specific, the representative of the collected samples, the completion and precision of recorded data, and the timely manner of data sharing and availability can ameliorate the output from computational modeling [19], [130], [131].”

Line 414f.: Please consider adding that timely availability of sequence data is a third missing issue. Please see, doi: 10.1016/j.tim.2017.09.001 and doi: 10.1038/s41598-017-18791-z

Thank you for the references. We have included this point (the sentence below and line 490 in the revised version) and added several sentences to elaborate the issue in lines 508-513. 

“However, three main limitations exist in current surveillance: 1) the imbalanced sampling efforts on different hosts and geographical regions; 2) the incompletion of data records [130]; and 3) the delayed availability of sequence data [19], [131].”

“GISAID [35]and GenBank [36], theseopen access database platforms have facilitated the accessibility and sharing of influenza sequence data to the science community. Despite the availability of these platforms, the sharing of viral sequence data is often long after the outbreaks and records are frequently incomplete [137]. Therefore, a standardized protocol on how to record collected samples and what information is needed to report should be established for sharing more complete viral and host-related information.”

Line 416f.: How should such policies look like? Please describe/explain.

The ideal policies would fully rely on the representative of viral samples with high quality of epidemiological studies to allocate resource. But the reality is that for example we have more samples from the U.S. than other countries and regions, which results in a biased database. 

To clarify, we have revised the sentence as follows (Lines 492-493).

“Policies to globally optimize resource allocation with considering the representative of collected samples from outbreaks in different regions are needed.”

Line 417f.: Why is the imbalance sampling for zoonotic influenza more severe? Please explain.

The unequal sampling for zoonotic influenza is more severe, because samples from animals are mostly case-based, that is, only severe cases have the probability to be sampled. But for avian influenza, the non-symptomatic co-existing of viruses and hosts produces viral genetic diversity and the potential pandemic isolate from reassortments of multi-subtype co-infections (https://www.ncbi.nlm.nih.gov/pmc/articles/PMC5119462/). Therefore, planned sampling in zoonotic influenza may help the prediction of emerging influenza viruses. 

Line 419: Replace “zoonosis” with “zoonotic”. Or what are you trying to say here?

Done.

Line 420: The sentence “which are recognized as the natural reservoir of avian influenza A viruses (AIV) [111]” should go to the introduction.

Agreed. We have moved this sentence to introduction (Lines 101-104).

“The seasonal vaccines offer a little or no protection to emerging zoonotic influenza viruses with pandemic potential, as many species, especially wild aquatic birds, are recognized as the natural reservoir of all subtypes of influenza A viruses and have the potential to occur spillover and infect humans directly [20].”

Line 425f.: Please explain why “a sample-based accumulation curve” can help in identifying virus subtypes.

A sample-based accumulation curve can provide an initial rationalization and optimal surveillance strategy based on identifying a subtype-specific percentage of influenza virus in circulation in a given time interval. It has the advantages to “estimate the virus subtype diversity once sampling is underway and then optimize the sampling strategy to maximize subtype detection while minimizing samples to be collected and tested”. This method tries to capture the genetic diversity with the most cost-effective way for surveillance. Details can be found (https://journals.plos.org/plosone/article?id=10.1371/journal.pone.0090826)

Line 428: Why is there an issue with data records, please explain 

This paragraph talks about missing information in data records. To avoid confusion, we have revised it as follows (Lines 504-508)

The affiliated sequence meta-data records have been improved with samples from recent years. But the epidemiological information, viral phenotypic characteristics and host characteristics are not sufficiently recorded. With no accurate information on geographical region, host species and migratory pathways and viral characteristics, we do lose lots of power in our model inference [136], not to say improving the prediction of viral evolvability.”

Line 433: Why and how would such a protocol help? Please explain.

The standardized protocol proposed here is to set a uniform data collection and submission standards for influenza surveillance. For example, the protocol provides guidance on how to record collected samples and what the minimum of viral and host-related information should be reported. 

Line 436: I do not see how and where in the previous section your statement has been shown/explained.

Sorry for the confusion, we have revised this sentence as below and in lines 515-516 in the revised version.

“As shown in Figure 2, computational models that fully integrate multiple sources of information, including experimental evidence, could aid in the identification of critical components for vaccine design.”

Line 437: How do we benefit from more advances in computer power? Please explain.

The rapid improvement on computing capability has allowed us to apply models to a much larger dataset and to perform develop more expansive models that capture more biological reality which could not have been analyzed previously. For example, the phylodynamic modeling can efficiently handle hundreds of genetic sequences with a large dimension of the trait states. Another example is the structured coalescent now comes to more use with incorporating population demographic to infer viral dynamics. This has been included in this current revision (Lines 522-526)

“More complicated and realistic models previously limited by computing capability can be developed with the advances of computing power [138]. For example, it becomes possible to develop viral phylodynamic models that can incorporate results from laboratory experiments of viral antigens and host immune responses[108]; The development on structured coalescent for better estimation on viral population and mutation or migration events [139], [140].”

Line 438f.: What would be the effect of such a viral phylodynamic model?

It can explore the interactions of ecological, epidemiological and immunological factors that potentially determine viral evolution and diversity. The brief summary of viral phylodynamic models are in the part “Pathogen Evolvability”. 

Line 443: Why is this the ultimate goal? Machine learning is barely mentioned before. Please explain in detail how it can help. 

Thank you for catching this point. We briefly introduced machine learning in the next paragraph. It can help to integrate different models and conduct optimization based on input data and automatic iterations of simulations. To avoid confusion, we have revised “ultimate goal” to a more reasonable description as “the next step” at lines 529-530 in the revised version.

“With all these,the next stepwould be to introduce and apply machine learning to the computational process for vaccine design. “

Line 462: Definition of genetic drift is missing. Same for antigenic drift, mentioned before. However antigenic shift is not mentioned at all.

Thank you. We have changed the sentence as below and lines 547 in the revised version.

“The rapid genetic changes and antigenic driftof influenza virus populations…”

With antigenic draft, we now defined it in lines 84-86 in the revised version as below. 

“The HA, however, undergoes rapid antigenic drift that accumulates from point mutations under immune selection pressure in the major antigenic sites, allowing the virus to escape neutralizing antibody responses [12]and resulting in imprecise prediction of circulating strains.”

In this review, we did not specifically talk about the term “antigenic shift”, but referred to as “zoonotic strains” that infect humans. The purpose is to include not only the strains that caused pandemic but also the strains that infect humans directly, for example, H5N1 and H7N9. 

Reviewer 3 Report

This is a very well-written review focused on cutting-edge computational-based approaches to identify conserved antigens and predict optimal B and T cell epitopes for universal influenza vaccine design. The figures are excellent and the scope is an accurate coverage of the primary strategies employed for vaccine discovery. There are a few points that need additional clarification and some additional minor corrections are noted.

1.    Part 2 (Line 99): This is an excellent conceptual overview but it would benefit from providing additional specific examples for each design approach. A table summarizing each strategy, examples of one or more vaccines employing this strategy under development and advantages/drawbacks of each would improve the paper.

2.    Table 1 should include a column that lists what organization or company is developing the indicated vaccine

3.    How the antigens or epitopes are presented as an immunogen or their structure is an important part of vaccine design and computational design tools are making significant advances in this area. The authors briefly mention VLPs. Additional discussion on this strategy and other approaches (i.e. nanoparticles) where computational design tools are employed to design an optimal scaffold or framework to present conserved epitopes or antigens is needed.

4.    What is the potential impact of computational modeling of the immune response data generated by testing new universal influenza vaccines? This review focuses on epitope/antigen discovery but not all predicted designs will be successful. Some additional perspective on how computational tools can be further employed to refine vaccine design is needed.

Minor

Line 105, no comma after escape

Line 106, hemagglutinin assay should be singular not plural

Line 111, Delete “a” (in amajor informational gaps)

Lines 122-123, check this sentence – it needs a verb

Line 125, is > are

Line 314, showed > shown

Line 380, phenotypic > phenotype

Author Response

Reviewer 3: Comments and Suggestions for Authors
This is a very well-written review focused on cutting-edge computational-based approaches to identify conserved antigens and predict optimal B and T cell epitopes for universal influenza vaccine design. The figures are excellent and the scope is an accurate coverage of the primary strategies employed for vaccine discovery. There are a few points that need additional clarification and some additional minor corrections are noted.

Thank you for the useful feedback and detailed comments. 

1.    Part 2 (Line 99): This is an excellent conceptual overview but it would benefit from providing additional specific examples for each design approach. A table summarizing each strategy, examples of one or more vaccines employing this strategy under development and advantages/drawbacks of each would improve the paper.

Thank you for the valuable comment. We have highlighted each approach, including advantages and disadvantages in Table 1 and Table 2. Together these tables provide a brief summary of preclinical and clinical examples. Table 1 is new addition to the review manuscript (shown below). 

2.    Table 1 should include a column that lists what organization or company is developing the indicated vaccine

Thank you. We have added a new column that provides information on the organization or company is developing vaccine. Please see below revised table, which is Table 2 in the revised manuscript. 

3.    How the antigens or epitopes are presented as an immunogen or their structure is an important part of vaccine design and computational design tools are making significant advances in this area. The authors briefly mention VLPs. 
Additional discussion on this strategy and other approaches (i.e. nanoparticles) where computational design tools are employed to design an optimal scaffold or framework to present conserved epitopes or antigens is needed.

Thank you for the comments. We briefly discuss the challenge of epitope presentation, specifically we discuss epitope grafting in epitope-based vaccine design. This has been added to the text at lines 269 -280 and is copied below.

“A major challenge in the design of epitope-based vaccines is to focus immune response onto multiple well-conserved epitopes in order to elicit broad protective/neutralizing immune responses. Epitope grafting or scaffolding, has been proposed as a solution for epitope-based vaccine design. In this method, minimal epitopes that are highly conserved in pathogen are grafted onto an appropriate heterologous-protein scaffold. Approaches for scaffold selection and design include single algorithm-based tools like MAMMOTH or meta-servers like TM-align and consensus-based designs [30]. Three main criteria have been proposed for the selection of scaffold that include size, where smaller-sized scaffolds help to focus immune responses to grafted epitopes while preventing unwanted responses to scaffold. Second criterion is the flexibility of scaffold with a possible positive correlation between flexibility and immunogenicity. The third criterion is the structural environment of the graft. A well-defined structural boundary between protein scaffold and epitopes enhances the specificity of immune responses [30].”

4.    What is the potential impact of computational modeling of the immune response data generated by testing new universal influenza vaccines? This review focuses on epitope/antigen discovery but not all predicted designs will be successful. Some additional perspective on how computational tools can be further employed to refine vaccine design is needed.

We agreed that computational approaches alone will not be sufficient to develop a universal vaccine. In this revision we discuss the need for experimental evidence to complement computational tools as shown in Figure 1. We added new paragraph in “3.2 Integration of experimental evidence and model development” (shown below) to address the importance of testing the predicted design and how to use experimental data to improve the modeling. 

“3.2 Integration of experimental evidence and model development

As shown in Figure 2, these computational models could efficiently compute and select critical components for vaccine design. However, we cannot solely rely on computational design, where computed antigens have uncertain biological effects. Experimental evidence (Figure 1) from animal models or approved human clinical trials are valuable to be incorporated into computational design. The experimental data on pathogen immunogenicity and host immune system can first provide preliminary evidence on natural or computed antigens and further amplify the usage of this new evidence to the computational procedure for more accurate prediction and evaluation [19].”

Minor

Line 105, no comma after escape

Corrected as follows

While global surveillance efforts and data sharing agreements have increased available information, vaccine design often ignores the underlying processes of the global influenza meta-population which generates diversity that allows the viral populations to escape vaccine-induced immune responses and anti-viral treatments.”

Line 106, hemagglutinin assay should be singular not plural

Done

Line 111, Delete “a” (in amajor informational gaps)

Done

Lines 122-123, check this sentence – it needs a verb

Done

Line 125, is > are

Done

Line 314, showed > shown

Done

Line 380, phenotypic > phenotype

Done